# Seroprevalence and Factors Associated with Scrub Typhus Infection among Forestry Workers in National Park Offices in South Korea

**DOI:** 10.3390/ijerph18063131

**Published:** 2021-03-18

**Authors:** Ji-Hyuk Park, Byoungchul Gill, Dilaram Acharya, Seok-Ju Yoo, Kwan Lee, Jeongmin Lee

**Affiliations:** 1Department of Preventive Medicine, College of Medicine, Dongguk University, Gyeongju 38066, Korea; dilaramacharya123@gmail.com (D.A.); medhippo@dongguk.ac.kr (S.-J.Y.); kwanyia@dongguk.ac.kr (K.L.); 2Division of Bacterial Diseases, Korea Disease Control and Prevention Agency, Cheongju 28159, Korea; gilri@korea.kr; 3Division of Healthcare Technology Development, Ministry of Health and Welfare, Sejong 30113, Korea

**Keywords:** scrub typhus, seroepidemiologic survey, forestry workers

## Abstract

Scrub typhus is caused by the arthropod-borne bacterium *Orientia tsutsugamushi* and is an endemic infectious disease in the Asia-Pacific area. This study aimed to investigate the seroprevalence of scrub typhus and identify associated risk and protective factors among forestry workers, a neglected risk group for scrub typhus, in National Park Offices in South Korea. A nationwide cross-sectional serosurvey was carried out on 1945 National Park Office forestry workers (NPOFWs) in South Korea during December 2016. We visited 29 main offices and used a structured questionnaire to collect data regarding general characteristics, work activities, work hygiene-related factors, and other potential risk factors. Serum samples from NPOFWs were tested using indirect immunofluorescence assay to detect *O. tsutsugamushi* immunoglobulin (Ig) G and M antibodies. Of the 1945 NPOFWs, 718 (36.9%) participated in this cross-sectional study. The seroprevalence, defined as ≥1:256 for IgG and/or ≥1:16 for IgM, was 4.9% (35/718). In multivariate logistic analysis, longer duration of work in national parks (≥15 years; odds ratio (OR), 4.19; 95% confidence interval (CI), 1.71–10.28) and dry field farming (OR, 2.47; 95% CI, 1.12–5.46) were significantly associated with a higher risk of scrub typhus infection. Furthermore, the risk of scrub typhus infection was significantly lower among NPOFWs who washed working clothes daily (OR, 0.37; 95% CI, 0.18–0.75). This study indicated that scrub typhus is an important disease among NPOFWs in South Korea. Work hygiene, especially washing working clothes daily, needs to be emphasized among NPOFWs. Additionally, more precautions are required to diminish the rate of scrub typhus infection among NPOFWs who perform dry field farming.

## 1. Introduction

Scrub typhus, also known as tsutsugamushi disease, is a febrile infectious illness caused by the arthropod-borne bacterium *Orientia tsutsugamushi* [1]. Approximately 1 to 3 weeks after being bitten by infected larval trombiculid mites (chiggers), scrub typhus may induce fever, rash, eschar at the bite site, headache, myalgia, and lymphadenopathy. Although most symptoms are mild, severe complications such as multiorgan failure, encephalitis, interstitial pneumonia, and myocarditis have been reported [2]. Appropriate antibiotics such as tetracycline and doxycycline provide effective treatments, however no licensed vaccines are available to prevent scrub typhus [3].

Growing numbers of scrub typhus cases have been reported in many countries with more than 1 million cases occurring annually worldwide [4]. The disease has been endemic in the area known as the tsutsugamushi triangle, which extends from northern Japan and far-eastern Russia in the north, to northern Australia in the south, and Pakistan in the west [1]. Furthermore, the burden of scrub typhus has extended from the tsutsugamushi triangle and its genetic diversity demonstrates it remains a global disease, which is supported by a recent report of a novel species of *Orientia chuto* acquired by a patient in Dubai, another divergent *Orientia* in a Chilean patient, and a serologically diagnosed case in Africa [5,6,7].

In South Korea, scrub typhus remains endemic and has been a national notifiable infectious disease since 1994 [8]. The incidence of scrub typhus has been showing a decreasing trend over recent years as indicated by an annual incidence of 21.52 (11,105 cases) in 2016 and 7.73 (4005 cases) per 100,000 population in 2019 [9]. The fall is the main season for contracting scrub typhus in South Korea, and *Leptotrombidium scutellare* and *L. pallidum* are the predominant transmission vectors in South Korea [8]. Several serologic studies [10,11,12] and case-control studies [13,14,15] on scrub typhus have been conducted on different study populations in South Korea. However, to the best of researchers’ awareness, none of the studies has assessed the seroprevalence and risk factors of scrub typhus among forestry workers.

Seroepidemiologic studies among forestry workers, one of the neglected risk groups for scrub typhus, from nationally representative data would enable the identification of risk factors and provide evidence to support the development and implementation of meaningful public health measures both nationally and in other endemic countries. As such, we aimed to determine the seroprevalence and identify the risk and protective factors of scrub typhus infection among forestry workers in the National Park Office in South Korea.

## 2. Materials and Methods

### 2.1. Study Population

In South Korea, a total of 22 areas have been designated as national parks. The Korea National Park Service manages 21 of these national parks, except Hallasan National Park on Jeju Island. The National Park Offices (NPOs) are executive organs of the Korea National Park Service and focus on the management of park resources to ensure professional and scientific management and the provision of high-quality tourist services. According to data obtained from the Korea National Park Service, 1945 National Park Office forestry workers (NPOFWs, 1150 at 29 main offices and 795 at 65 branch offices) were based at NPO offices in August 2016 and 718 participated in this cross-sectional study.

### 2.2. Data Collection

A structured questionnaire was developed based on the results of a literature review and a preliminary meeting with several NPOFWs. The questionnaire addressed general characteristics, work activities, work hygiene-related factors, and other potential risk factors, including additional jobs (including rice and dry field farming) and raising animals. Five study teams visited the 29 main NPOs during 21–30 December 2016 (winter season in South Korea). NPOFWs in branch offices were asked to visit the nearest main offices on appointed dates. We administered the questionnaire and collected a blood sample (10 mL) from each participant.

### 2.3. Serologic Testing

Blood samples were centrifuged immediately and serum samples were sent to the Korea Center for Disease Control and Prevention to test for scrub typhus. An indirect immunofluorescence assay (IFA), the standard method of serologic diagnosis of scrub typhus [16], was used to detect *O. tsutsugamushi* (Gilliam, Karp, Kato, and Boryong) antibody immunoglobulin G (IgG) and immunoglobulin M (IgM). Samples were initially screened at 1:16 and titrated to 1:2048 when seroreactive. Antibody titers of ≥1:256 for IgG and/or ≥1:16 for IgM were considered seropositive, as these are the criteria used to diagnose scrub typhus in South Korea [17].

### 2.4. Statistical Analysis

Statistical analysis was carried out using SPSS version 20.0 (SPSS, IBM, Armonk, NY, USA). Univariate logistic regression analysis was conducted to assess associations between potential risk and protective factors and seroprevalence of scrub typhus among NPOFWs. All variables with a significance level of <0.10 by univariate logistic regression analysis were entered to multivariate logistic regression analysis with backward elimination to calculate odds ratios (ORs) and 95% confidence intervals (CIs). Statistical significance was accepted for *p*-values < 0.05.

### 2.5. Ethics

The Institutional Review Board of Dongguk University Gyeongju Hospital reviewed and approved the study protocol before the seroepidemiologic survey (approval number: 16–286). The aims and objectives of the study were explained before enrollment to all participants who provided written and informed consent.

## 3. Results

### 3.1. Demographic Characteristics and Serologic Results

Of the 1945 NPOFWs in South Korea, 718 participated in this study with a response rate of 36.9% (main NPOFWs: 40.5%, branch NPOFWs: 31.7%). The participants consisted of 545 men (75.9%) and 173 women (24.1%). The mean age and duration of work were 43.3 (range, 18–71) years and 8.8 (range, 0.2–38.0) years, respectively. Thirty-five of the 718 participants (4.9%) were seropositive. Titer cutoffs for IgG antibodies against *O. tsutsugamushi* ranged from <1:16 to >1:2048; 6 samples (0.8%) had IgG titers of ≥1:256. Titer cutoffs for IgM antibodies against O. tsutsugamushi ranged from <1:16 to 1:128; 31 samples (4.3%) had IgM titers ≥1:16. Seropositivity for both IgG and IgM was found in 2 samples (0.3%, Table 1). All seropositive participants were healthy and experienced no symptoms of scrub typhus.

### 3.2. Univariate Analysis of Scrub Typhus Seroprevalence and Potential Risk Factors

Longer duration of work (≥15 years) was associated with a higher risk of scrub typhus infection compared to shorter duration of work (<5 years) (*p* = 0.001). However, age, sex, region, organization type, and level of education were not associated with scrub typhus seroprevalence (Table 2). Work activities, including monitoring of natural resources, grass mowing, and cleaning, were not found to be associated with scrub typhus seroprevalence (Table 3). The questionnaire also contained information about the use of personal protective equipment, risky and protective behaviors during outdoor work, and precautionary activities after outdoor work. Only washing working clothes daily after outdoor work had a lower risk of scrub typhus infection (*p* = 0.009, Table 4). In addition, among other potential risk factors, dry field farming was associated with a higher risk of scrub typhus infection (*p* = 0.013, Table 5).

### 3.3. Multivariate Analysis of Scrub Typhus Seroprevalence and Potential Risk Factors

We included the following factors in the multivariate model based on a *p*-value < 0.10 by univariate logistic regression analysis: duration of work, washing working clothes daily after work, and dry field farming. After backward stepwise regression, all variables remained in the model. Longer duration of work (≥15 years, OR, 4.19; 95% CI, 1.71–10.28) and dry field farming (OR, 2.47; 95% CI, 1.12–5.46) were significantly associated with a higher risk of scrub typhus infection. Furthermore, the risk of scrub typhus infection was significantly lower among NPOFWs who washed working clothes daily (OR, 0.37; 95% CI, 0.18–0.75, Table 6).

## 4. Discussion

This study documents the seroprevalence rate of scrub typhus and its associated risk and protective factors based on an analysis of nationwide data obtained from NPOFWs in South Korea. We found that the seroprevalence via IFA was 4.9% (IgG titer ≥1:256 and/or IgM titer ≥1:16). Other South Korean studies conducted on non-symptomatic individuals living in areas with a high incidence of scrub typhus have reported seroprevalence rates of 3.4–9.7% for all ages and 1.7–3.8% for those <60 years old [10,11,12]. In the present study, the seroprevalence rate was slightly higher among NPOFWs (mean age, 43.3 years) than for non-symptomatic individuals aged <60 years.

The seroprevalence rates of scrub typhus vary by country, target population, serologic methods, and their criteria used for the assessment [18]. We were unable to find another IFA-based serologic study on scrub typhus among forestry workers. An IFA-based Sri Lankan study reported a seroprevalence rate of 2.0% (IgG titer ≥1:128) among military workers, which was similar to that observed among NPOFWs in the present study (1.8%) [19]. However, an IFA-based nationwide study among healthy people in Bhutan found a considerably higher seroprevalence rate of 22.6% (IgG titer ≥1:256 or IgM titer ≥1:1024) [20], which probably reflects a higher incidence of scrub typhus in Bhutan (62 cases per 100,000 in 2015) [21] than in South Korea (18.5 cases per 100,000 in 2015) [9].

We found that NPOFWs who had worked for longer periods of time had a significantly higher risk of scrub typhus infection, presumably because they were more likely to be exposed to *O. tsutsugamushi*-infected mites. However, specific types of work activities and work hygiene-related factors during outdoor work, except washing working clothes daily, were not significantly associated with scrub typhus seroprevalence. Although a previous community-based case-control study performed in South Korea reported several risk factors for scrub typhus (taking a rest on the grass, working in the short sleeves, working with bare hands, and squatting to defecate/urinate) [14], another South Korean case-control study found no association between such outdoor work hygiene-related factors and scrub typhus seroprevalence [13]. Sample size and different ages and residency in cases might influence these results.

Multivariate logistic regression analysis also identified that washing working clothes daily after outdoor work was significantly associated with a lower risk of scrub typhus infection. Another case-control study performed in India reported that the risk of scrub typhus was significantly higher among individuals who did not change clothes or undergarments after outdoor work [22], while two Korean case-control studies reported that changing a uniform after outdoor work was not significantly associated with the risk of scrub typhus [13,14]. However, a literature search failed to identify any study on the association between washing working clothes daily and scrub typhus infection. Nevertheless, washing could remove infected mites from working clothes, and thus, reduce the risk of exposure to *O. tsutsugamushi*.

Farmers and forestry workers are at high-risk of scrub typhus infection [3], and in the present study, NPOFWs that performed dry field farming were more likely to be seropositive, whereas those involved in rice farming were not. Similarly, a case-control study on scrub typhus in South Korea reported a significant association with dry field farming and no association with rice farming. In South Korea, dry field farming is performed near mountains, whereas rice farming is performed near rivers usually in low-lying areas [14]. The risk of scrub typhus infection was lower among NPOFWs who used insect repellents without statistical significance. However, considering that insect repellents are used generally to avoid scrub typhus infection [23], the relatively low proportion of using insect repellents (17.8%) among NPOFWs might need to be improved.

In South Korea, this was the first nationwide seroepidemiologic study of scrub typhus infection to be conducted among forestry workers. However, the findings in this study have several limitations. First, response rates were slightly different between main and branch office NPOFWs, which was probably caused by our visiting main NPO offices, and this may have affected seroprevalence rates. Second, scrub typhus seroprevalence in this study may have been influenced by seasonality because we collected serum samples during December, just after the peak period (from October to November) for scrub typhus in South Korea [24]. Third, NPOFWs on Jeju Island in South Korea were not included in this study.

## 5. Conclusions

The seroprevalence rate of scrub typhus among South Korean NPOFWs was found to be 4.9%. Longer duration of work and dry field farming were identified as significant risk factors, while washing working clothes daily had a significant protective effect. Thus, washing working clothes daily needs to be emphasized to diminish scrub typhus infection among NPOFWs. Furthermore, if NPOFWs perform dry field farming, suitable precautions should be taken.

## Figures and Tables

**Table 1 ijerph-18-03131-t001:** Serologic results for *Orientia tsutsugamushi* antigen among National Park Office forestry workers in South Korea.

Titer	IgG		IgM	
	No.	%	No.	%
<1:16	667	92.9	687	95.7
1:16	13	1.8	18	2.5
1:32	17	2.4	8	1.1
1:64	8	1.1	3	0.4
1:128	7	1.0	2	0.3
≥1:256	6	0.8	0	0.0
Total	718	100.0	718	100.0

Ig, immunoglobulin.

**Table 2 ijerph-18-03131-t002:** Association between demographic characteristics and scrub typhus seroprevalence among National Park Office forestry workers in South Korea.

Variables	Total	Seroprevalence No. (%)	OR (95% CI)	*p* Value ^a^
Sex				
Men	545	27 (5.0)	1.08 (0.48–2.41)	0.861
Women	173	8 (4.6)	Reference	
Age (years)				
<29	109	3 (2.8)	Reference	
30–49	366	16 (4.4)	1.62 (0.46–5.65)	0.453
≥50	243	16 (6.6)	2.49 (0.71–8.73)	0.154
Duration of work (years)				
<5	295	8 (2.7)	Reference	
5–<15	288	12 (4.2)	1.56 (0.63–3.87)	0.338
≥15	135	15 (11.1)	4.48 (1.85–10.86)	0.001
Region				
Northeastern	229	14 (6.1)	Reference	
Southwestern	489	21 (4.3)	0.69 (0.34–1.38)	0.294
Organization types				
Main National Park Offices	466	20 (4.3)	Reference	
Branch National Park Offices	252	15 (6.0)	1.41 (0.71–2.81)	0.326
Education				
High school or less	239	16 (6.7)	1.73 (0.87–3.44)	0.115
University or more	478	19 (4.0)	Reference	

OR, odds ratio; CI, confidence interval. ^a^ Univariate logistic regression was applied.

**Table 3 ijerph-18-03131-t003:** Association between work activities and scrub typhus seroprevalence among National Park Office forestry workers in South Korea.

Variables	Total	Seroprevalence No. (%)	OR (95% CI)	*p* Value ^a^
Monitoring of natural resources				
Yes	301	11 (3.7)	0.62 (0.30–1.29)	0.198
No	416	24 (5.8)	Reference	
Repairing facilities				
Yes	464	26 (5.6)	1.61 (0.74–3.49)	0.228
No	253	9 (3.6)	Reference	
Supervision of illegal activities				
Yes	427	23 (5.4)	1.32 (0.65–2.69)	0.448
No	290	12 (4.1)	Reference	
Patrolling				
Yes	595	31 (5.2)	1.62 (0.56–4.68)	0.372
No	122	4 (3.3)	Reference	
Guiding visitors				
Yes	445	25 (5.6)	1.57 (0.74–3.31)	0.241
No	273	10 (3.7)	Reference	
Exploration program				
Yes	174	7 (4.0)	0.77 (0.33–1.80)	0.550
No	544	28 (5.1)	Reference	
Grass mowing				
Yes	324	15 (4.6)	0.91 (0.46–1.80)	0.776
No	393	20 (5.1)	Reference	
Cleaning				
Yes	366	21 (5.7)	1.47 (0.73–2.93)	0.280
No	351	14 (4.0)	Reference	

OR, odds ratio; CI, confidence interval. ^a^ Univariate logistic regression was applied.

**Table 4 ijerph-18-03131-t004:** Association between work hygiene-related factors and scrub typhus seroprevalence among National Park Office forestry workers in South Korea.

Variables	Total	Seroprevalence No. (%)	OR (95% CI)	*p* Value ^a^
During outdoor work				
Wearing a long-sleeved shirt				
Yes	601	30 (5.0)	1.17 (0.44–3.07)	0.755
No	116	5 (4.3)	Reference	
Wearing long pants				
Yes	681	35 (5.1)	NA	0.998
No	37	0 (0.0)	Reference	
Wearing gloves				
Yes	602	29 (4.8)	0.92 (0.37–2.27)	0.855
No	115	6 (5.2)	Reference	
Wearing boots				
Yes	577	28 (4.9)	1.14 (0.46–2.81)	0.777
No	140	6 (4.3)	Reference	
Wearing a hat				
Yes	537	25 (4.7)	0.92 (0.42–2.01)	0.839
No	179	9 (5.0)	Reference	
Using insect repellents				
Yes	127	4 (3.1)	0.63 (0.22–1.82)	0.389
No	588	29 (4.9)	Reference	
Resting on the grass				
Yes	117	6 (5.1)	1.07 (0.43–2.63)	0.889
No	601	29 (4.8)	Reference	
Using a mat to rest				
Yes	196	10 (5.1)	1.06 (0.50–2.26)	0.871
No	520	25 (4.8)	Reference	
Eating meals in woodland				
Yes	180	6 (3.3)	0.60 (0.25–1.48)	0.270
No	537	29 (5.4)	Reference	
Defecating/urinating in woodland				
Yes	64	4 (6.3)	1.34 (0.46–3.92)	0.594
No	654	31 (4.7)	Reference	
After outdoor work				
Taking a shower				
Yes	612	28 (4.6)	0.68 (0.29–1.59)	0.373
No	106	7 (6.6)	Reference	
Taking a bath				
Yes	341	16 (4.7)	0.98 (0.49–1.95)	0.952
No	376	18 (4.8)	Reference	
Changing working clothes daily				
Yes	526	24 (4.6)	0.79 (0.38–1.64)	0.521
No	192	11 (5.7)	Reference	
Washing working clothes daily				
Yes	546	20 (3.7)	0.40 (0.20–0.80)	0.009
No	172	15 (8.7)	Reference	

OR, odds ratio; CI, confidence interval; NA, not available. ^a^ Univariate logistic regression was applied.

**Table 5 ijerph-18-03131-t005:** Association between other potential factors and scrub typhus seroprevalence among National Park Office forestry workers in South Korea.

Variables	Total	Seroprevalence No. (%)	OR (95% CI)	*p* Value ^a^
Additional jobs				
Rice farming				
Yes	22	1 (4.5)	0.93 (0.12–7.10)	0.942
No	696	34 (4.9)	Reference	
Dry field farming				
Yes	100	10 (10.0)	2.64 (1.23–5.67)	0.013
No	618	25 (4.0)	Reference	
Orchard farming				
Yes	26	2 (7.7)	1.66 (0.38–7.34)	0.501
No	692	33 (4.8)	Reference	
Livestock farming				
Yes	16	0 (0.0)	NA	0.999
No	702	35 (5.0)	Reference	
Raising animals				
Dogs (outside homes)				
Yes	78	3 (3.8)	0.76 (0.23–2.54)	0.656
No	640	32 (5.0)	Reference	
Dogs (inside homes)				
Yes	52	4 (7.7)	1.71 (0.58–5.04)	0.333
No	666	31 (4.7)	Reference	
Cats				
Yes	32	0 (0.0)	NA	0.998
No	686	35 (5.1)	Reference	
Recognition of scrub typhus				
Yes	680	31 (4.6)	0.41 (0.14–1.22)	0.107
No	38	4 (10.5)	Reference	

OR, odds ratio; CI, confidence interval; NA, not available. ^a^ Univariate logistic regression was applied.

**Table 6 ijerph-18-03131-t006:** Multivariate logistic regression analysis of important variables (*p* < 0.10) associated with scrub typhus seroprevalence among National Park Office forestry workers in South Korea.

Variables	OR (95% CI)	*p* Value
Duration of work (years)		
<5	Reference	
5– < 15	1.41 (0.56–3.52)	0.467
≥15	4.19 (1.71–10.28)	0.002
Washing working clothes daily		
Yes	0.37 (0.18–0.75)	0.006
No	Reference	
Dry field farming		
Yes	2.47 (1.12–5.46)	0.025
No	Reference	

OR, odds ratio; CI, confidence interval.

## Data Availability

The dataset analyzed during the current study are available from the corresponding author on reasonable request.

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
