# Peer review of "Seroprevalence and Factors Associated with Scrub Typhus Infection among Forestry Workers in National Park Offices in South Korea"

_ijerph, 2021, doi:10.3390/ijerph18063131_

Round 1
Reviewer 1 Report
In this study, the authors performed the seroprevalence of scrub typhus infection among NPOFWs in South Korea. This seroepidemiologic survey revealed a rate of 4.9% seropositive IgG and/or IgM, and further uncovered that longer duration of work, dry field farming and washing working clothes daily as associated factors with scrub typhus infection among NPOFWs. The work was well-designed and logically explanation with a strictly statistical analysis. Although the sample size is small, the conclusion is reliable and could be considerable to a valuable prospect in the prevention of scrub typhus epidemiology. There are still some concerns to explain or clarify.
(1) About the study population, did all the NPOFWs represent the regional distribution in South Korea? If yes, the authors could attempt to mark these regions in map with the signs of forest.
(2) In addition, considering the seasonal prevalence of scrub typhus caused by the arthropod-borne vector, the season or period of data collection needs to indicate clearly.
Author Response
Thank you for reviewing this article and giving us careful comments. We have revised the manuscript based on the given comments. All changes have been indicated with red colored writings in the manuscript.
Point 1: About the study population, did all the NPOFWs represent the regional distribution in South Korea? If yes, the authors could attempt to mark these regions in map with the signs of forest.
Response 1: The sample size is small to represent the regional distribution and we could not include the NPOFWs on Jeju Isalnd in South Korea. We added this limitation in the discussion (Line 216-217).
Point 2: In addition, considering the seasonal prevalence of scrub typhus caused by the arthropod-borne vector, the season or period of data collection needs to indicate clearly.
Response 2: We indicated the season and period of data collection according to your comment (Line 85-86, Line 213-216).
Reviewer 2 Report
The work presented in this paper seeks to determine the seroprevalence of Orientia tsutsugamushi in National Parks Office Forestry workers in South Korea. Indirect immunofluorescent assay was used to detect serum immunoglobulin G and M, respectively against O. tsutsugamushi. Overall, the manuscript is worthy of publication. The following comments should be addressed.
Major comments: The authors should discuss the lack of correlation between seroprevalence of O. tsutsugamushi and work activities. This finding is different from previous reports. What could explain the difference between the results from the study mentioned in reference 14?
Minor comments: The authors should carefully review the manuscript to correct the grammatical errors and typos found in the Abstract.
Author Response
Thank you for reviewing this article and giving us careful comments. We have revised the manuscript based on the given comments. All changes have been indicated with red colored writings in the manuscript.
Point 1: The authors should discuss the lack of correlation between seroprevalence of O. tsutsugamushi and work activities. This finding is different from previous reports. What could explain the difference between the results from the study mentioned in reference 14?
Response 1: We think that sample size and different ages and residency in cases might influence these results. We added the sentence according to your comment (Line 187-188).
Point 2: The authors should carefully review the manuscript to correct the grammatical errors and typos found in the Abstract.
Response 2: We carefully reviewed and corrected the manuscript according to your comment (Line 15-33).